# Junk DNA Hypothesis: A Task-Centric Angle of LLM Pre-trained Weights through Sparsity

## Abstract

The traditional notion of "Junk DNA" has long been linked to non-coding segments within the human genome, constituting roughly 98% of its composition. Initially perceived as biologically inert, recent research has unveiled the critical roles some of these seemingly non-functional DNA sequences play in cellular processes. Intriguingly, the weights within deep neural networks exhibit a remarkable similarity to the redundancy observed in human genes. It was believed that weights in gigantic models contained excessive redundancy, leading to the conception that a significant number of parameters could be removed without compromising performance.

This paper challenges this conventional wisdom by presenting a compelling **counter-argument**. We employ sparsity (specifically weight pruning) as a tool to isolate and quantify the nuanced significance of low-magnitude weights in pre-trained large language models (LLMs). Our study demonstrates a strong correlation between these weight magnitudes and the knowledge they encapsulate, from a downstream **task-centric** angle. Drawing parallels with biological insights, we raise the **"Junk DNA Hypothesis"** backed by our in-depth investigation: *while small-magnitude weights may appear "useless" for simple tasks and suitable for pruning, they actually encode crucial knowledge necessary for solving more difficult downstream tasks. Removing these seemingly insignificant weights can lead to* *irreversible* *knowledge forgetting and performance damage in difficult tasks.*

To study it formally, we introduce several quantifiable metrics for gauging **downstream task difficulty**: (i) within the same task category, we vary the adequacy of target domain data (e.g., few-shot fine-tuning) and extend this to multi-domain learning (e.g., majority versus minority language in multilingual translation). Additionally, we assess the availability of external information (e.g., open-book versus close-book QA); (ii) across diverse task categories, we utilize the normalized performance gap between humans and models as an indicator of LLM-facing task complexity. Our extensive experiments validate the Junk DNA Hypothesis across a spectrum of model scales, tasks, and datasets, employing both forms of sparsity - unstructured and structured (N:M). We further empirically confirm that the essential knowledge indeed resides within the pre-trained weights, and the performance drop does not stem from constrained model capacity post-pruning. These findings offer fresh insights into how LLMs encode knowledge in a task-sensitive manner, pave future research direction in model pruning, and open avenues for task-aware conditional computation during inference. Codes will be released.

## 1 Introduction

The human genome, an astonishing compilation of three billion DNA base pairs, reveals a fascinating dichotomy. Approximately 2% of this vast genetic landscape encodes proteins, leaving the remaining portion seemingly superfluous (Carey, 2015). This non-coding section of the genome has earned the moniker "Junk DNA" (Ohno, 1972), positing that large genomes would inevitably harbor non-coding sequences, passively accumulated over millennia, devoid of any protein-coding capacity. Yet over the past decade, this notion was challenged it has become evident that at least some of these seemingly extraneous DNAs play essential roles in cellular function. For example, these regions of DNA contain vital sequences that act as regulatory elements (Zheng et al., 2010). Specialized proteins known as transcription factors bind to these elements, orchestrating the intricate dance of gene transcription,

either activating or repressing it. This revelation underscores the intricate and functional complexity of what was once considered "Junk DNA", revealing previously hidden layers of genetic regulation.

In a parallel vein, a prevailing belief in the realm of deep neural networks bears resemblance to the notion of Junk DNA in biology. This belief suggests that a substantial portion of parameters, particularly those with low magnitude, within deep neural networks lack significance and can be vastly pruned across various architectures and applications (Han et al., 2016; Gale et al., 2019; Frankle & Carbin, 2019; Mocanu et al., 2018; Liu et al., 2022b; Chen et al., 2020). Moreover, as model sizes continue to expand, the volume of redundant parameters is poised to escalate (Liu et al., 2022a). This principle extends its effectiveness even to billion-level Large Language Models (LLMs) (Jaiswal et al., 2023a; Frantar & Alistarh, 2023; Sun et al., 2023). With the negligible loss in performance stemming from the absence of low-magnitude weights, a widely held belief has taken root: these small-magnitude weights are essentially superfluous components that make scant contribution to the model's functionality. At this juncture, it behooves us to pause and pose a question: *Could there be crucial facets overlooked in the context of whether these low-magnitude weights are truly inconsequential artifacts for large-scale models?*

This paper addresses the aforementioned query by employing (mainly) magnitude-based pruning to discern and quantify the subtle importance of low-magnitude weights. Magnitude-based pruning, a well-established method known for consistently producing robust results, has been extensively utilized (Han et al., 2016; Frankle & Carbin, 2019; Liu et al., 2022b; Fernandez-Lopez et al., 2023; Ma et al., 2023). Yet, it is imperative to clarify that this paper **does not aim to be another LLM pruning** exposition. Instead, we see pruning as a *quantitative and easily controllable tool* for probing and comprehending the role of small-magnitude weights in pre-trained LLMs. The **crux of our research** lies in adopting a novel **task-centric** viewpoint towards pre-trained weights. In other words, these weight magnitudes and the information they embody exhibit a significant correlation with the complexity of the downstream task for which the pre-trained LLM will be employed. Our study disrupts conventional assumptions by providing a **counterpoint** regarding the previously disregarded yet pivotal function of small-magnitude weights. Drawing a parallel with biological insights, we articulate our discoveries as the **Junk DNA Hypothesis**, stated as follows:

> **Junk DNA Hypothesis (pertaining to LLMs):** While small-magnitude weights might seem nearly superfluous for simple downstream tasks and thus candidates for pruning, they actually encode vital knowledge essential for tackling more challenging downstream tasks. Removing these ostensibly inconsequential weights can result in irreparable loss of knowledge and performance degradation in difficult tasks.

The primary challenge in formalizing this conjecture lies in providing a precise, controllable **definition of "task difficulty"**. We undertake this exploration through the following avenues:

- **Within the Same Task Category**: we first vary the adequacy of target domain data (Liu et al., 2019) (e.g., few-shot fine-tuning), and we then extend this idea to multi-domain learning (e.g., majority versus minority language in multilingual translation (Liu et al., 2020)). Additionally, we investigate the influence of external information availability, as exemplified by open-book versus closed-book QA (Ram et al., 2023).

- **Across Diverse Task Categories**: for each task, we utilize the gap between the best human performance, and the target LLM model's performance on the task (normalized by human performance), as an indicator of task complexity "sensed" by that LLM. Such complexities can be compared across tasks for the same LLM.

Our extensive experiments substantiate the Junk DNA Hypothesis across a diverse range of model sizes, tasks, and datasets, employing both unstructured and structured sparsity (N:M). While the **overarching notion** that "more challenging downstream tasks permit less room for pruning" may not come as a surprise, our study unveils **several subtler, often unexpected findings**:

- Moving beyond the nebulous distinction between simple and complex tasks, the various conceptions of task difficulty defined above by us, both within and across tasks, appear to align closely with the behavior of pruning fragility. This suggests practical methods for estimating the task-dependent achievable degree of LLM sparsity. In certain tasks, even a

modest reduction in low-magnitude pre-trained weights (e.g., 10%) results in a significant drop in accuracy, underscoring their pivotal role in effectively handling more intricate tasks.

- We confirm that for difficult tasks, the essential knowledge indeed resides within the pre-trained weight values. Our carefully designed experiments (e.g., dense versus sparse fine-tuning after pruning pre-trained weights) demonstrate that the decline in performance does not originate from limited model capacity post-pruning (e.g., see Figure 6). Conversely, if we freeze all pre-trained weights and only update as few as 10% of the largest-magnitude ones, we can often match the performance of fully fine-tuning dense models.

- Junk DNA Hypothesis holds true when transitioning from unstructured to structured N:M pruning. Counter intuitively, N:M sparsity consistently outperforms unstructured sparsity at very high levels of sparsity, possibly because it avoids layer collapse (e.g., see Figure 4-b).

## 2 RELATED WORK

### 2.1 CLASSICAL PRUNING AND SPARSE NEURAL NETWORKS

Pruning removes specific parts of a deep neural network, such as weights, neurons, or filters. The initial purpose of pruning is retrospectively to accelerate the model at inference time (a.k.a., post-training sparsification (Mozer & Smolensky, 1989; LeCun et al., 1990). Post-training sparsification has been well studied and results in various mature criteria that can be generally categorized into zero-order methods (Han et al., 2016; Gale et al., 2019), first-order methods (Molchanov et al., 2016; Sanh et al., 2020; Jiang et al., 2021), and second-order methods (LeCun et al., 1990; Hassibi & Stork, 1992; Dong et al., 2017) - the last usually achieve higher performance but also are more expensive due to the Hessian calculation, leading to the development of many Hessian approximation approaches (Zeng & Urtasun, 2018; Wang et al., 2019; Singh & Alistarh, 2020; Kurtic et al., 2022). The Lottery Ticket Hypothesis (LTH) (Frankle & Carbin, 2019) utilizes iterative magnitude pruning (IMP) to identify a subnetwork at initialization that can be re-trained independently to the original dense network's performance. Sparse training (Mocanu et al., 2018; Jaiswal et al., 2022; Mostafa & Wang, 2019; Evci et al., 2020; Liu et al., 2021; Yuan et al., 2021; Yin et al., 2023; Kundu et al., 2021), on the other hand, starts with a (random) sparse network and updates network connectivity during training to search for good sparse neural network without any pre-training and dense training steps.

### 2.2 SPARSITY IN LARGE-SCALE MODELS

The advent of large-scale pre-trained models has led to the development of advanced post-training pruning methods, aiming to enhance the cost-effectiveness of these expansive models (Sanh et al., 2020; Chen et al., 2020; Jaiswal et al., 2023b; Zafrir et al., 2021; Kurtic et al., 2022; Xu et al., 2021; Lagunas et al., 2021; Zhang et al., 2022; Frantar et al., 2021; Jaiswal et al., 2023a; Ma et al., 2023). Among them, Frantar et al. (2021) extend second-order pruning to the BERT-level scale, enabling the pruning of blocks of weights and achieving state-of-the-art results for sparse BERT. Frantar & Alistarh (2023) introduce SparseGPT for pruning large language models (LLMs) in a single shot without requiring re-training or fine-tuning. They leverage column-wise second-order pruning, and successfully remove 100B weights from OPT-175B without a significant increase in perplexity. More recently, Sun et al. (2023) propose a straightforward pruning method that takes both weights and activations into account, demonstrating comparable performance to Frantar & Alistarh (2023). Li et al. (2022) reveal that activation sparsity is a prevalent phenomenon in Transformers (90% of intermediate output), yielding another opportunity for acceleration. Liu et al. (2023) introduce a large-scale SMC-Bench, indicating that state-of-the-art magnitude- and/or gradient-based sparse algorithms fall short when applied out-of-the-box to larger-scale models and a selected of complex downstream tasks. Our study is inspired by Liu et al. (2023), but with significantly expanded experiment scales, versatile task choices, concrete task difficulty definitions, and richer insights.

## 3 EXPERIMENT WITHIN THE SAME TASK CATEGORY

In this section, we will evaluate Junk DNA Hypothesis within the same task category, by defining task difficulties concretely in three different settings. We will specifically validate if the pre-trained values are the true gems through carefully designed control experiments.

### 3.1 THREE DIFFICULTY SETTINGS WITHIN THE SAME TASK

❶ **Task Difficulty Setting 1:** *Varying the Adequacy of Target Domain Data*

**Rationale:** The difficulty of learning a task is commonly thought to be influenced by the number of available training examples: fewer data points typically imply more challenges to learn well. To quantitatively control task difficulty within a single task, we manually manipulate the *volume of data used for fine-tuning* by randomly sampling various ratios from the target domain dataset. This allows us to *disentangle task difficulty from the task type*.

**Method:** To examine the influence of small-magnitude weights, we conduct a comparative analysis between two models: one starting from the pre-trained model with small-magnitude weights, and the other without. The former is commonly referred to as task-specific fine-tuning on downstream tasks, denoted as **Dense Transfer** in this paper. The latter model, named **Sparse Transfer**, differs from dense transfer in the way that we first perform magnitude pruning on the pre-trained model, creating a sparse model. We then fine-tune on downstream tasks while keeping the sparse mask fixed.

For pruning, we employ the widely adopted one-shot magnitude-based pruning (Han et al., 2016), a basic approach that removes small-magnitude weights from a pre-trained network. We consider two types of sparsities: (1) **Unstructured Sparsity**: individual weights in the model are zeroed out independently, leading to irregular zero patterns (LeCun et al., 1990; Han et al., 2016); (2) **Structured $N$:$M$ Sparsity**: a fine-grained sparsity pattern in which only $N$ weights are non-zero for every continuous $M$ weights (Nvidia, 2020; Zhou et al., 2021). We report the results of $M$=8 ($N$ ranges from 7 to 1) in the main paper and leave those of $M$=4 ($N$ ranges from 3 to 1) in Appendix A.

Many advanced techniques, such as iterative pruning with re-training (Frankle & Carbin, 2019), second-order pruning (Kurtic et al., 2022), learning rate rewinding (Renda et al., 2020), and knowledge distillation (Hinton et al., 2015), can all yield stronger empirical performance for sparse transfer (Liu et al., 2023). However, we intentionally opt for one-shot magnitude pruning in order to isolate the effect of small-magnitude weights as the sole "delta" between the two models, and due to its recently observed promising performance on large language models (Jaiswal et al., 2023a). For $N$:$M$ sparsity, we similarly perform magnitude-based single-shot pruning as described in Zhou et al. (2021).

**Model and Datasets:** In this setting, we test the Junk DNA hypothesis with the pre-trained RoBERTa-Large/Base models (Devlin et al., 2018) provided by Hugging Face[1]. We choose the downstream tasks of SST-2, QNLI, MNLI from the classical GLUE benchmark (Wang et al., 2018). Following common practice, we do not prune the embedding layer nor the classifier. Furthermore, we use the same fine-tuning recipe for both dense and sparse models, ensuring a fair comparison between them.

❷ **Task Difficulty Setting 2:** *Majority v.s. Minority in Multi-Domain Learning*

**Rationale:** Setting 2 essentially extends Setting 1 (more or less data, in a single domain) to a multi-domain scenario (Daras et al., 2022): assuming multiple data domains will be involved in the downstream task, we conjecture the data-rich domain is easier to learn, than the data-scarce domain. Specifically, we focus our study on a multilingual translation task with majority versus minority languages and consider *translation across majority language pairs (i.e., those with ample pretraining or fine-tuning data) is categorized as an easy task*. Conversely, *translation across minority language pairs (i.e., "low-resource" ones at pretraining/fine-tuning) is considered as a hard task*.

**Method:** Similar to the Setting 1, we will compare **Dense Transfer** and **Sparse Transfer** methods. Our evaluation comprises two distinct regimes: (i) In the "*zero-shot*" scenario, we perform language translation tasks without any fine-tuning. The Dense/Sparse models are directly assessed without any form of "transfer". In this context, the task's complexity (i.e., whether the languages involved are categorized as majority or minority) is determined by the volume of training data available for the respective languages during the pre-training phase; (ii) In the "*few-shot*" scenario, the dense/sparse models undergo fine-tuning on specific language pairs before being evaluated on the language translation task involving those particular languages. In this situation, the task's difficulty is defined by the amount of data employed for those languages during the fine-tuning phase.

**Model and Datasets:** We choose the official mBART model (Liu et al., 2020) for multilingual translation. This model was initially pre-trained on 25 languages using the masked language modeling

---

[1]https://huggingface.co/docs/transformers/model_doc/roberta

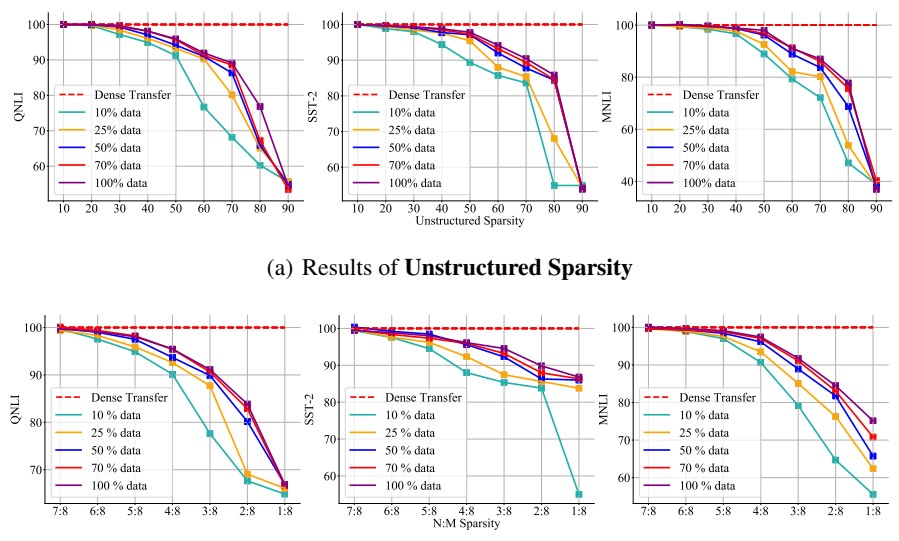

(a) Results of **Unstructured Sparsity**

(b) Results of **N:M Sparsity** (M=8, N ranges from 1 to 7)

Figure 1: **Varying target domain data adequacy**: Dense Transfer vs. Sparse Transfer using RoBERTa-Base on various downstream tasks. Each sub-figure showcases a specific downstream task, with various % of data volume. Task difficulty is measured by the training data volume. Note that in the figures, the performance of sparse transfer is **normalized** by the dense transfer performance.

(MLM) approach. From the original pre-training set, we narrow down to a subset of 10 languages, and continue training mBART on language pairs from this selected subset, following Tang et al. (2020) with default configurations of 40K iterations and Adam optimizer at a learning rate of $10^{-6}$.

In the downstream, we assess mBART by selecting four languages from the open-source parallel corpus (OPU, 2020). Following the methodology described by Arivazhagan et al. (2019), we utilize pivot data via English to establish 2-to-2 majority language translation pairs (Ru,Vi) and 2-to-2 minority language translation pairs (Gu, My). The statistics of these four languages in the pre-training dataset, CC-25 (Liu et al., 2020), is presented in the Appendix Table 2.

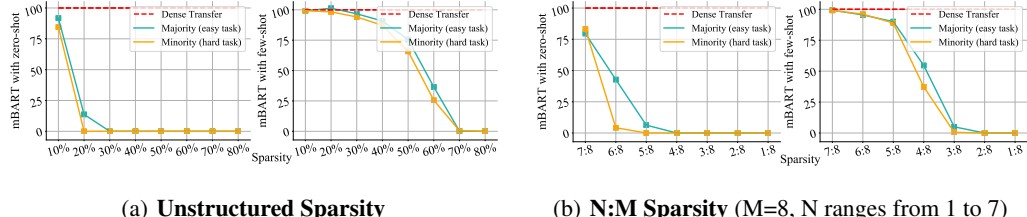

(a) **Unstructured Sparsity**     (b) **N:M Sparsity** (M=8, N ranges from 1 to 7)

Figure 2: **Majority vs minority languages**: Results on mBART with multilingual language translation. In either figure, the left plot is the "*zero-shot*" evaluation and the right "*few-shot*" evaluation. In both figures, the performance of sparse transfer is normalized by the dense transfer performance.

❸ **Task Difficulty Setting 3:** *With v.s. Without Available External Information*

**Rationale:** Setting 3 posits that a task aided by external information will be inherently easier to solve in comparison to the same task undertaken without such external support. Our focus is specifically on the Question-Answering (QA) task, where we compare two distinct settings: (i) *Open-book QA*, which permits the model to consult and extract information from external sources; and (ii) *Closed-book QA*, where the model must rely solely on its internal knowledge. We postulate that Open-book QA represents an easier task when compared to Closed-book QA.

**Method:** Similar to the previous two settings, we will juxtapose the **Dense Transfer** and **Sparse Transfer** methodologies. It is worth noting that in the creation of the sparse pre-trained model for Setting 3, we not only utilize magnitude-based pruning, but also incorporate two other latest LLM pruning techniques: SparseGPT (Frantar & Alistarh, 2023) and Wanda (Sun et al., 2023). While these

two methods do not exactly remove elements based on magnitude, our aim is to ascertain whether the Junk DNA hypothesis can be extended to encompass other weight importance criteria as well.

**Model and Datasets:** We choose Vicuna-7B (Chiang et al., 2023), an open-source chatbot trained by fine-tuning LLaMA on user-shared conversations collected from ShareGPT. For this task setting, we use a popular reading comprehension dataset, TriviaQA (Joshi et al., 2017), which includes 95K question-answer pairs authored by trivia enthusiasts and independently gathered evidence documents, six per question on average, that provide high-quality distant supervision for answering.

TriviaQA consists of fairly complex composition based questions with considerable syntactic and lexical variability between questions and corresponding answer-evidence sentences, making it a challenging enough test-bed for our evaluation. We access Vicuna-7B performance by conditioning questions with and without evidence sentences in open and closed book setting respectively.

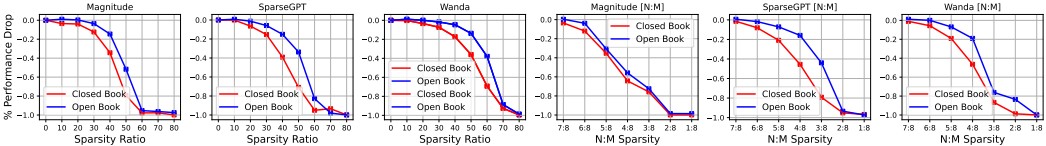

Figure 3: **Open-book vs Close-book QA**: Results on Vicuna-7B with sparse models generated by magnitude pruning (left figure), SparseGPT (middle), and Wanda (right). In all figures, the performance of sparse transfer is normalized by the dense transfer performance.

## 3.2 MAIN RESULTS: VALIDATING THE JUNK DNA HYPOTHESIS

We present our findings from Settings 1-3, in Figures 1, 2, and 3, respectively. These results provide robust support for our Junk DNA hypothesis, and we summarize the key observations below:

① **Removal of small-magnitude weights is viable to some extent for easier tasks:** In all three settings involving easy tasks (i.e., 70% and 100% data volume in Setting 1, and Open-book QA in Setting 3), we find that it is feasible to discard 30%-50% of small-magnitude weights at once without compromising performance in unstructured sparsity. This indicates that for simple tasks, the knowledge encoded in high-magnitude weights is sufficient to handle the task. Consequently, fine-tuning the high-magnitude weights proves to be adequate, rendering small weights unnecessary. We note that this conclusion aligns with the prior finding in Jaiswal et al. (2023a).

② **Eliminating small weights leads to irreversible performance degradation in more challenging tasks:** In stark contrast to the ease with ample data volume in Setting 1, fine-tuning with limited data, such as 10% and 25% of the original dataset, can only achieve performance parity with dense models at a maximum sparsity level of 20%. In Setting 2, we have observed an even more remarkable phenomenon: the removal of just 5% of weights on the harder task (i.e., translation across minority languages), leads to a noticeable decline in performance for both "zero-shot" and "few-shot" evaluations. It is noteworthy that the performance degradation caused by pruning small-magnitude weights is more pronounced in the "few-shot" setting compared to the "zero-shot" setting, indicating the greater importance of small weights when the target domain data is accessible. Furthermore, a similar trend is observed in the Close-book task in Setting 3, where models experience performance deterioration even at trivial sparsity levels, such as 5% to 10%, when small-pruned weights are eliminated. Overall, the above results show that those "useless" small weights are imperative to encode crucial knowledge necessary for solving more challenging downstream tasks.

③ **Junk DNA persists in both N:M sparsity and unstructured sparsity, and beyond the magnitude criteria**. Both N:M sparsity and unstructured sparsity yield similar observations, indicating the presence of Junk DNA in both settings. We highlight a counter-intuitive discovery in this context: Despite a more constrained pattern, N:M sparsity consistently outperforms unstructured sparsity at extremely high levels of sparsity, highlighted in Figure 3. This advantage is likely attributed to its inherent ability to prevent layer collapse, which commonly happens for global magnitude pruning.

We also evaluate Junk DNA Hypothesis with **two non-magnitude pruning methods**: SparseGPT and Wanda. We apply them to Vicuna-7B and evaluate the sparse models on Open-book QA and Closed-book QA. A similar performance trend can be observed across all pruning methods as shown

in Figure 3: While Wanda appears to perform slightly better than magnitude pruning and SparseGPT, all of them consistently demonstrate the same trend: suffering from larger performance degradation on Close-book QA which is more challenging than Open-book QA. Therefore, we conclude that Junk DNA Hypothesis can be extended to encompass other weight importance criteria as well.

## 4  EXPERIMENT ACROSS DIVERSE TASK CATEGORIES

In this section, we broaden our Junk DNA hypothesis to encompass a range of task categories. However, quantifying and comparing task difficulty across diverse task types poses a significant challenge due to a multitude of contributing factors. These factors include data type, size, distribution, task definition, loss function, and more. The primary research question at hand is to identify a dependable task difficulty estimator that remains agnostic to the aforementioned factors.

❹ **Task Difficulty Setting 4:** *Estimating LLM-facing Task Difficulty by Normalized Human-LLM Performance Gap*

**Rationale and Method:** We propose a method to gauge complexity by juxtaposing the performance of deep learning models with that of human counterparts. Specifically, we define task difficulty as the disparity in performance between humans and models, normalized by human performance. A more pronounced positive performance gap (for instance, where humans outperform the machine to a greater extent) would signify a higher level of difficulty for the model in handling the given task. Conversely, in cases where the machine outperforms humans, a larger gap indicates an easier task[2]. The resulting assessment of across-task difficulty is outlined in Table 1.

Specifically, we choose a range of downstream tasks, including SST-2, COLA, QQP, STS-B, QNLI, MNLI, and RTE from the GLUE benchmark (Wang et al., 2018) conducted on RoBERTa Large and Base models. Additionally, we incorporate tasks involving commonsense reasoning (CSQA, WinoGrande) as well as arithmetic reasoning (MAWPS, SVAMP) from the SMC-Bench (Liu et al., 2023). It is worth noting that certain metrics used for comparison may not align perfectly (e.g., SST-2 vs. COLA; QQP vs. STS-B), which may introduce some limitations in our comparisons. However, we emphasize that our assessment of task difficulty aligns well with intuitions derived from previous studies (Wasserblat, 2021; Ko & Choi, 2020). These studies suggest that COLA is more challenging than SST-2, and STS-B is more difficult than QQP, respectively.

**Result:** The findings depicted in Figure 4 echo the conclusions drawn in Section 3, once more providing robust support for the validity of the *Junk DNA hypothesis across a broad spectrum of task categories*. While it may be feasible to remove small-magnitude weights without significant repercussions in simpler tasks, these pre-trained small weights contain vital downstream knowledge essential for tackling more difficult tasks, and thus are no longer dispensable.

Table 1: Measuring the Across-Task Difficulty by the Performance Difference between humans and models (normalized by human performance): the larger (positive) margin, the more difficult for the machine. Human performance is obtained from Nangia & Bowman (2019). The more challenging task is marked in bold.

| | Single Sentence | | Sentence Similarity | | Natural Language Inference | | Commonsense Reasoning | |
| --- | --- | --- | --- | --- | --- | --- | --- | --- |
| | SST-2 | COLA | QQP | STS-B | QNLI | RTE | WinoGrande | CSQA |
| Human | 97.8 | 66.4 | 80.4 | 92.7 | 91.2 | 93.6 | 94.0 | 89.0 |
| RoBERTa-Large | 96.2 | 64.9 | 91.8 | 92.2 | 94.4 | 84.3 | 78.1 | 72.1 |
| "Task Difficulty" (%) | 1.64 | **2.26** | -14.18 | **0.54** | -3.51 | **9.94** | 16.91 | **18.99** |

## 5  ARE PRE-TRAINED MAGNITUDE VALUES INDEED THE TRUE GEM?

Having recognized the pivotal role of small weights in downstream adaptation, particularly in relation to task difficulty, our next objective is to delve into the foundational factors contributing to the crucial function of small weights during fine-tuning. Our primary research inquiries are outlined below:

• *Which holds greater significance: the knowledge (weight values) stored in pre-trained small-magnitude weights, or the potential to adjust these weights through fine-tuning?*

---

[2]We acknowledge the limitations in our approach, such as when both humans and LLMs perform poorly on a task, potentially indicating high difficulty not reflected in our 'relative' gap metric. We aim to inspire further research in accurately assessing cross-task difficulty

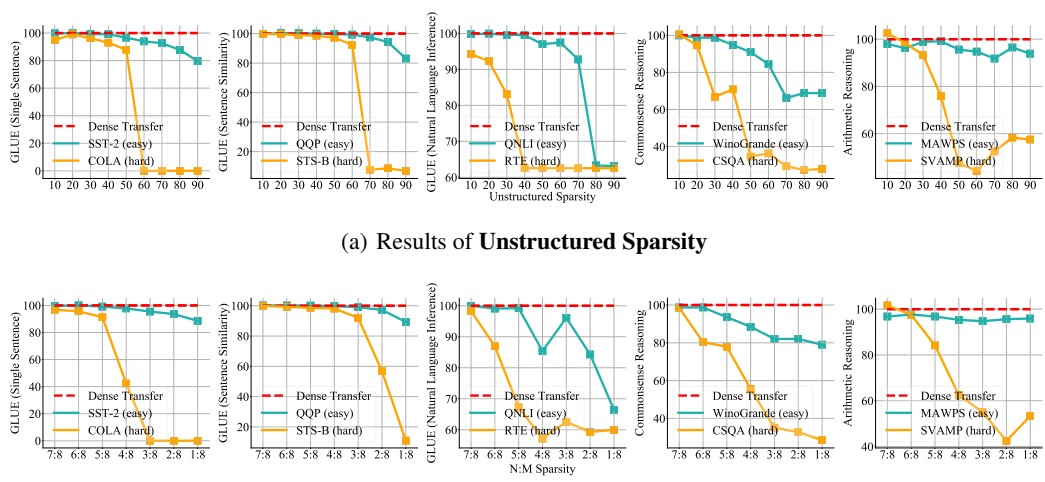

(a) Results of **Unstructured Sparsity**

(b) Results of **N:M Sparsity** (M=8, N ranges from 7 to 1)

Figure 4: **Across-Task Difficulty via Normalized Human-LLM Performance Gap**: Dense Transfer vs. Sparse Transfer using RoBERTa-Large on various downstream tasks. Each sub-figure compares an easier task and a more challenging one. Note that in all figures, the performance of sparse transfer is normalized by the dense transfer performance.

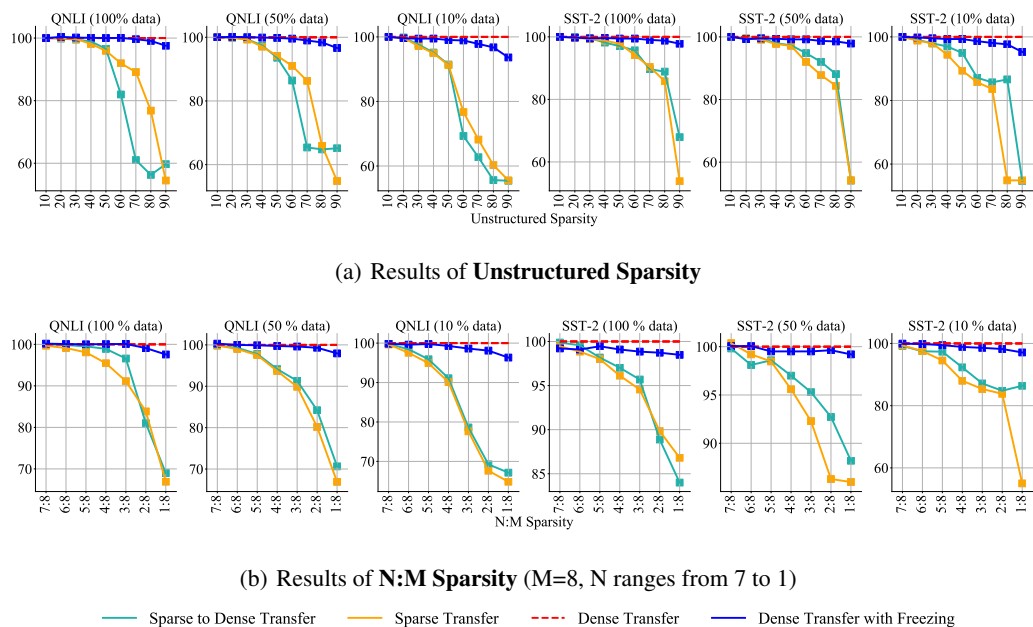

(a) Results of **Unstructured Sparsity**

(b) Results of **N:M Sparsity** (M=8, N ranges from 7 to 1)

Figure 5: **Varying target domain data adequacy**: four different fine-tuning settings with RoBERTa-Base on various downstream tasks. All performance is normalized by the one of Dense Transfer.

• *Is it possible to recover the knowledge embedded in pre-trained small-magnitude weights if we prune them and allow them to grow freely during fine-tuning?*

**Method:** To address these inquiries, we explore four comparison methods: (1) **Dense Transfer**: as described in Section 3.1; (2) **Dense Transfer with (Partial) Freezing**: a dense model where small-magnitude weights remain fixed during fine-tuning; (3) **Sparse Transfer**: as in Section 3.1; (4) **Sparse to Dense Transfer**: small-magnitude weights are initially pruned after pre-training, and subsequently during fine-tuning, those pruned weights are allowed to gradually regain non-zero values. This approach also aids in determining whether the knowledge within small-magnitude pre-trained weights is essential for performance or if their adaptability during fine-tuning takes precedence. We pick Setting 1 (within-task) and Setting 4 (across-task), to report their performance of RoBERTa-Large, on MNLI, QNLI, SST-2, as well as CSQA and WinoGrande.

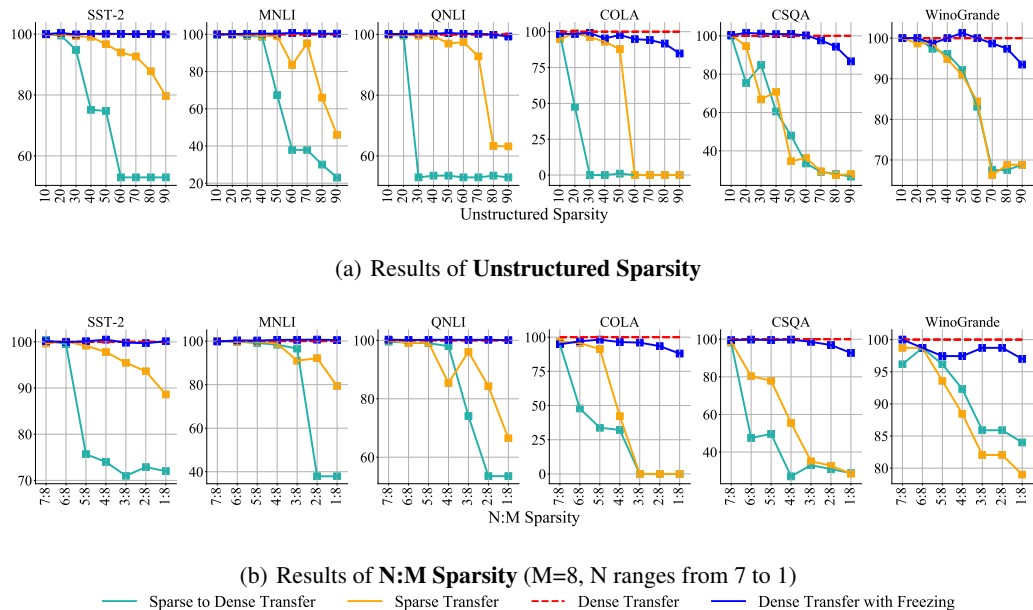

(a) Results of **Unstructured Sparsity**

(b) Results of **N:M Sparsity** (M=8, N ranges from 7 to 1)

Sparse to Dense Transfer    Sparse Transfer    Dense Transfer    Dense Transfer with Freezing

Figure 6: **Across-Task Difficulty via Normalized Human-LLM Performance Gap**: four different fine-tuning settings with RoBERTa-Large on various downstream tasks. Each sub-figure shows a specific downstream task. All performance is normalized by the one of Dense Transfer.

**Results:** The outcomes for both within-task difficulty and across-task difficulty are illustrated in Figure 5 and Figure 6, respectively. Below, we outline our key observations:

① **Pre-trained small weights harbor vital downstream knowledge, beyond mere free parameters.** Across both task-difficulty metrics, it becomes evident that settings preserving the pre-trained values of small weights—namely, Dense Transfer and Dense Transfer with Freezing—achieve superior performance when compared to the other two settings. The removal of small-magnitude weights from pre-trained models results in significant performance degradation, even when we permit the pruned weights to regenerate during fine-tuning. This observation strongly bolsters the Junk DNA Hypothesis, indicating that small-magnitude weights are far from redundant; rather, they house sophisticated knowledge crucial for downstream adaptation. This knowledge proves challenging to re-gain through fine-tuning, if these initial pre-trained weights are eliminated.

② **Freezing without updating yields commendable results.** Remarkably, on simpler tasks like SST-2, MNLI, and QNLI, maintaining an overwhelmingly large portion (90%) of small-magnitude weights in a frozen state leads to equally impressive performance without any loss. Even on more intricate tasks such as COLA, CSQA, and WinoGrande, freezing up to 70% of small-magnitude weights results in no discernible performance dip. This suggests that for easier tasks, the knowledge embedded in pre-trained small-magnitude weights is already more than sufficient. However, for more challenging tasks, allowing for moderate updates to all pre-trained weights remains essential.

# 6 CONCLUSION AND FUTURE WORK

In this study, we embark on an exploration to validate the prevailing belief that deep network weights are excessively redundant, allowing for a substantial pruning of parameters without compromising performance. Existing pruning algorithms typically operate under the assumption that low-magnitude weights are of limited significance and can be safely removed. However, our research presents a compelling counter-argument by unearthing the previously overlooked and intricate role of small-magnitude weights, closely tied to the difficulty level of downstream tasks. Through a comprehensive analysis of these low-magnitude weights, we make a significant revelation. Indeed, for certain straightforward tasks, these weights prove to be relatively "useless", making them suitable for pruning without adverse effects on performance. However, when it comes to tackling complex tasks, these small-magnitude weights carry crucial knowledge. Removing them in such scenarios can lead to irreparable damage to the performance of these challenging tasks. As the number of parameters in deep networks continues to grow exponentially, our findings prompt the exploration of directions such as task-complexity-dependent dynamic inference and network self-slimmable properties.

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

# A    RESULTS OF *N:M* SPARSITY WITH *M*=4

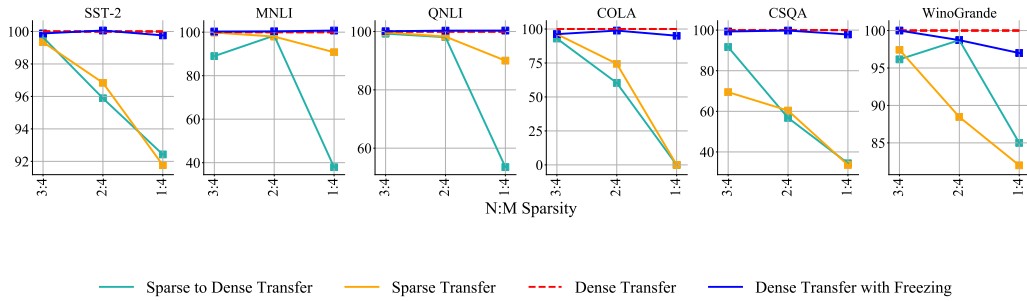

Figure 7: **Task Difficulty Setting 4:** Performance of four different fine-tuning settings with RoBERTa-Large on various downstream tasks. Each sub-figure showcases a specific downstream task. The across-task difficulty is justified by the performance gap between humans and models. All performance is normalized with the one of Dense Transfer.

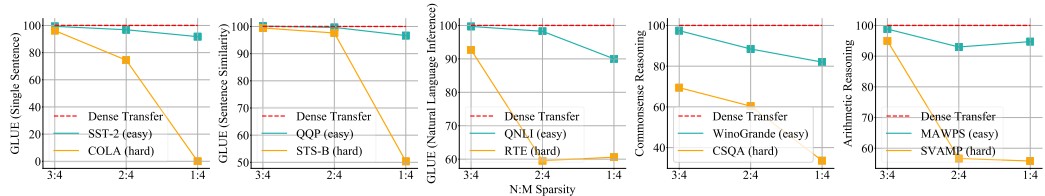

Figure 8: **Task Difficulty Setting 4:** Dense Transfer vs. Sparse Transfer using RoBERTa-Large on various downstream tasks. Each sub-figure showcases a specific downstream task, consisting of an easy dataset and a more challenging one. Across-task difficulty is justified by the performance gap between humans and models. Note that in the figures, the performance of sparse transfer is **normalized** by the dense transfer performance.

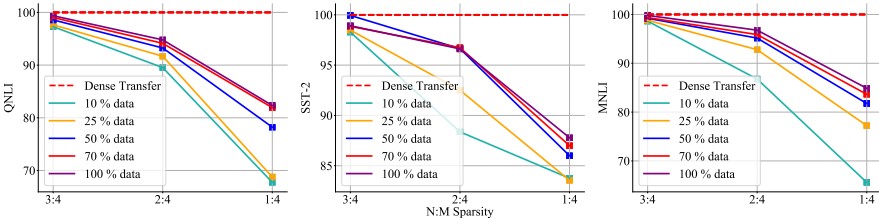

Figure 9: **Task Difficulty Setting 1:** Dense Transfer vs. Sparse Transfer using RoBERTa-Base on various downstream tasks. Sparse transfer involves pruning small-magnitude weights, which are then frozen as zeros during fine-tuning. Each sub-figure showcases a specific downstream task, with various % of data volume. Within-task difficulty is measured by the training data volume. Note that in the figures, the performance of sparse transfer is **normalized** by the dense transfer performance.

# B    SMALL-MAGNITUDE WEIGHTS CONTRIBUTE TO LOSS BASIN PRESERVATION

We also investigate the potential reasons behind the substantial performance drop resulting from the removal of small-magnitude weights on harder tasks. Our analysis revolves around the loss landscape, and we conjecture that small-magnitude weights play a significantly more crucial role in preserving the loss basin of the dense model on harder tasks compared to easier ones. Consequently, the absence of these small weights disrupts the optimal basin, leading to a considerable loss of performance.

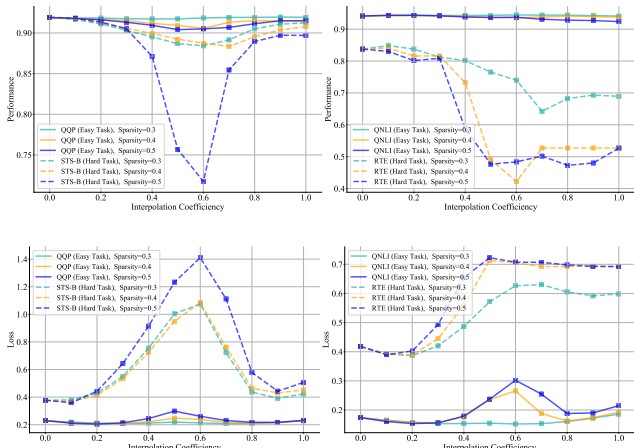

Figure 10: Linear interpolation from the Dense Transfer (**Left**) model to its corresponding Sparse Transfer models (**Right**) on easy and harder tasks (in terms of across-task difficulty).

To test our conjecture, we utilize the linear mode connectivity (LMC) metric proposed by (Frankle et al., 2020) between the solution produced by Sparse Transfer and that of Dense Transfer. Specifically, we perform linear interpolation between the fine-tuned model of Dense Transfer ($\boldsymbol{\theta}_d$) and the fine-tuned model of Sparse Transfer ($\boldsymbol{\theta}_s$), denoted as $\tilde{\boldsymbol{\theta}} = \alpha\boldsymbol{\theta}_s + (1 - \alpha)\boldsymbol{\theta}_d$. We conduct two sets of comparisons, namely QQP vs. STS-B and QNLI vs. RTE, and report the performance and loss in Figure 10. Our findings reveal that both sparse and dense models remain linearly connected, with minimal or no increase in loss barrier for easy tasks (i.e., QQP and RTE) when a certain portion of small-magnitude weights is removed. However, a significant increase in the loss barrier is observed when the same number of weights is removed for harder tasks. This observation strongly supports the concept of "Junk DNA", emphasizing the vital role of small-magnitude weights in ensuring that the fine-tuned model resides in the optimal basin. In contrast, the removal of these weights would lead to the destruction of this optimal basin, which is challenging to fix through fine-tuning, causing a notable decline in performance.

## C LANGUAGES AND STATISTICS FOR MULTILINGUAL TRANSLATION

Table 2: Languages and Statistics

| Language translation pairs | **Majority** | | **Minority** | |
|---|---|---|---|---|
| Code | Ru | Vi | Gu | My |
| Languages | Russian | Vietnamese | Gujarati | Burmese |
| Size/GB (CC25 for pre-training) | 278.0 | 137.3 | 1.9 | 1.6 |
| Size/MB (OPUS-100 for fine-tuning) | 116 | 42 | 22 | 2.5 |

