# OpenReview forum: "Junk DNA Hypothesis: A Task-Centric Angle of LLM Pre-trained Weights through Sparsity"
_ICLR.cc/2024/Conference — Submitted to ICLR 2024_

### Official Review · Reviewer_Q2er · 2023-10-30

**Soundness:** 3 good
**Presentation:** 3 good
**Contribution:** 3 good
**Rating:** 8
**Confidence:** 4

**Summary:**

This paper raises a very interesting argument, that the weights within deep neural networks exhibit a similarity to the redundancy observed in human genes in that they both contain seemingly non-functional elements that play a critical role in their respective systems. This similarity is due to the fact that low-magnitude weights in pre-trained LLMs may appear "useless" for simple tasks and suitable for pruning, but they actually encode crucial knowledge necessary for solving more difficult downstream tasks.

**Strengths:**

The main strength of this paper, compared to previous pruning works, is its task-centric viewpoint towards pre-trained weights. While it does not propose any new pruning way, the paper adopts a novel approach to isolate and quantify the significance of low-magnitude weights in pre-trained large language models (LLMs) by examining their correlation with the complexity of the downstream task for which the pre-trained LLM will be employed. This approach provides a more comprehensive understanding of the role of small-magnitude weights in LLMs and their impact on performance, particularly for complex tasks.

One of the main highlights of this paper is the authors' proposal of a method to quantitative define NLP downstream task difficulty. While this is in general highly ambiguous, the authors proposed (1) Varying the Adequacy of Target Domain Data; (2) Majority v.s. Minority in Multi-Lingual Translation, which essential extends the first setting to multi-domain learning; (3) QA with v.s. without available external Information, and (4) for different task types, as the disparity in performance between humans and models, normalized by human performance. The definition will be broadly useful for understanding LLM (both full and compressed) performance in fine granularity.

**Weaknesses:**

-	The fourth “cross-task difficulty” definition is flawed.  The authors assumed the larger the positive normalized performance gap between humans and models, the more difficult the task is for the model. However, if both human and LLM perform very poor (but “comparably poor”) on one task, it could mean this task is very difficult, yet in your setting the “relative” gap might not be significant. Besides, as the authors also pointed out, different tasks might have different metrics so directly normalizing and comparing across tasks can be problematic too.
-	It was known before difficult tasks are more fragile for pre-trained model pruning, such as in Sparsity-May-Cry (ICLR 2023). This paper essentially delves deeper on top of this exsiting observation.

**Questions:**

No particular question. The paper is very well written, and I enjoyed reading it. Great clarity and solidity, particularly in the way the authors organized their experiment.

---

> ### Author Response · Authors · 2023-11-21
> **Response to Reviewer Q2er**
>
> We sincerely thank you for your support and positive score. We are glad that you have found our definitions will be broadly useful for understanding LLMs. We now provide a detailed response to address the weaknesses that you have highlighted.
>
> **Question 1:** *The fourth “cross-task difficulty” definition is flawed. The authors assumed the larger the positive normalized performance gap between humans and models, the more difficult the task is for the model. However, if both human and LLM perform very poor (but “comparably poor”) on one task, it could mean this task is very difficult, yet in your setting the “relative” gap might not be significant. Besides, as the authors also pointed out, different tasks might have different metrics so directly normalizing and comparing across tasks can be problematic too.*
>
> - We appreciate your insightful critique of the “cross-task difficulty”. Your observation rightly points out a potential limitation in our methodology. Specifically, in cases where both humans and LLMs exhibit poor performance, the relative gap might not accurately reflect the true difficulty of the task. Nevertheless, the task pairs used in our paper were carefully selected by us, such that our assessment of task difficulty aligns well with intuitions derived from previous studies [1,2,3]. These studies suggest that COLA is more challenging than SST-2, STS-B is more difficult than QQP, and CSQA is more challenging than WinoGrande, respectively. Therefore, despite the limitation of our “cross-task difficulty” definition, we believe that our main conclusion remains valid within this context. We will highlight the current scope and limitations of the definition in the manuscript. We hope our work can inspire and encourage more efforts to accurately measure the cross-task difficulty.
>
>
> **Question 2:** *It was known before difficult tasks are more fragile for pre-trained model pruning, such as in Sparsity-May-Cry (ICLR 2023). This paper essentially delves deeper on top of this existing observation.*
>
> - While our study is indeed inspired by Sparsity-May-Cry (SMC), as rightly pointed by the reviewer, we delve deeper to study the role of small-magnitude weights in LLMs in addition to identifying the importance of the knowledge they encapsulate for downstream task performance. Specifically, compared with SMC, the unique contributions of the current draft can be summarized as follows:
>   1. Provided task criticality definitions: SMC estimates task difficulty mainly based on their ad-hoc results, while we provide more concrete measures (two large categories with four small categories) to identify task difficulty. Especially, the within task category is able to disentangle task difficulty from task type, providing a more convincing evaluation.
>   2. Exploring the effect of data volume on pruning.
>   3. Wider range of task choices: Small-scale: RoBERTa on various tasks from GLUE; Large-scale: billion-level models i.e., Vicuna-7B on Open-book and Close-book QA.
>   4. Exploring More sparsity patterns: we explored structured N:M sparsity.
>   5. Understand the importance of knowledge of small magnitude weights in the context of LLM fine-tuning.
>
> [1] Moshe Wasserblat. Best practices for text classification with distillation (part 2/4) – challenging use cases. https://www.linkedin.com/pulse/best-practices-text-classification-distillation-part-24-wasserblat/, 2021.
>
> [2] Bowon Ko and Ho-Jin Choi. Twice fine-tuning deep neural networks for paraphrase identification. Electronics Letters, 56(9):444–447, 2020.
>
> [3] Liu, Shiwei, Tianlong Chen, Zhenyu Zhang, Xuxi Chen, Tianjin Huang, Ajay Jaiswal, and Zhangyang Wang. "Sparsity May Cry: Let Us Fail (Current) Sparse Neural Networks Together!." arXiv preprint arXiv:2303.02141 (2023).

---

### Official Review · Reviewer_52Ea · 2023-11-01

**Soundness:** 3 good
**Presentation:** 3 good
**Contribution:** 2 fair
**Rating:** 6
**Confidence:** 2

**Summary:**

The paper focuses on the importance of the small weights in LLMs. They show that these are indispensable, particularly for harder tasks. The authors embrace a narrative to present similarities between the importance of these (previously thought to be "junk") weights and the junk DNA hypothesis in biology, which hypothesizes the unimportance of huge parts of DNA in humans for certain cellular processes and was proved to be wrong.

**Strengths:**

- The paper is well-written.
- The task-centric approach to the effects of small weights in LLMs is a good contribution to the AI community.
- The results of the paper are convincing.

**Weaknesses:**

- Most findings are not that surprising to me, for example, the finding that the small weights in LLMs can be important, or not, and that depends on the task. Nevertheless, this needs to be proved and this paper does it well.
- The paper could be improved if the error margins of the results were evaluated or included in the figures. If this might cause a huge additional computational burden (does it?), at least some statistical analysis of the significance of the results would help.

**Questions:**

- How do you explain the ups and downs in the figures? Specifically, for example in Fig.6a, the sparse-transfer 3:8 has better result than 4:8 in QNLI, and in Fig. 6b., sparse to dense transfer in CSQA 30% is higher than 20%, etc. Might such ups-and-downs indicate the variance of the results are high, and therefore the results are statistically insignificant?

---

> ### Author Response · Authors · 2023-11-21
> **Response to Reviewer 52Ea （1/2）**
>
> We sincerely thank the reviewer for the positive review. We are glad that you have found our paper to be impactful with important and convincing results! We now provide a detailed response to address the weaknesses that you have highlighted.
>
>
> **Question 1:** *Most findings are not that surprising to me, for example, the finding that the small weights in LLMs can be important, or not, and that depends on the task. Nevertheless, this needs to be proved and this paper does it well.*
>
> - We are grateful that you think our paper does well on verifying the Junk DNA hypothesis. While the importance of small-magnitude weights depending on task difficulty may not be very surprising, no previous work has comprehensively studied this point, likely due to the daunting challenges of quantifying task difficulty. While our paper does not introduce any new pruning method, we propose a novel way to quantify task difficulty including both within task and across task. Based on our approach, we provide a comprehensive understanding of the role of small-magnitude weights in LLMs, inspiring the exploration of directions such as task-complexity-dependent dynamic inference and network self-slimmable properties. As rightly highlighted by the **Reviewer Q2er**, quoting his/her comment:
>
>   > "One of the main highlights of this paper is the authors' proposal of a method to quantitatively define NLP downstream task difficulty. While this is in general highly ambiguous, the authors proposed (1) Varying the Adequacy of Target Domain Data; (2) Majority v.s. Minority in Multi-Lingual Translation, which essentially extends the first setting to multi-domain learning; (3) QA with v.s. without available external Information, and (4) for different task types, as the disparity in performance between humans and models, normalized by human performance. The definition will be broadly useful for understanding LLM (both full and compressed) performance in fine granularity."
>
>
> **Question 2:** *The paper could be improved by evaluating or including error margins in the results. If doing so is computationally burdensome, incorporating some statistical analysis to determine the significance of the results would be beneficial.*
>
> - Thank you for your valuable suggestion. We would like to clarify that the results presented in our submission are averages derived from three different random seeds. We chose not to include the error margin in the plot because the values represented are not original; they are after normalization based on the mean value of the dense transfer.
>
>   Based on your concerns, we take RoBERTa-Large on QNLI task as an example and conducted **20 runs** instead of our previous 3 runs. We report our mean and standard deviation as follows:
>
>   |                          | Dense Transfer   | 7:8           | 6:8           | 5:8           | 4:8           | 3:8           | 2:8           | 1:8          |
>   |--------------------------|------------------|---------------|---------------|---------------|---------------|---------------|---------------|--------------|
>   | Sparse to Dense Transfer | 0.91 ± 0.11      | 0.91 ± 0.11   | 0.85 ± 0.17   | 0.91 ± 0.11   | 0.87 ± 0.14   | 0.54 ± 0.1    | 0.51 ± 0.021  | 0.5 ± 0.0053 |
>   | Sparse Transfer          | 0.91 ± 0.11      | 0.91 ± 0.11   | 0.91 ± 0.11   | 0.91 ± 0.11   | 0.85 ± 0.17   | 0.86 ± 0.11   | 0.8 ± 0.094   | 0.62 ± 0.035 |
>   | Dense Transfer with Freezing | 0.91 ± 0.11 | 0.89 ± 0.15   | 0.95 ± 0.0021 | 0.94 ± 0.019  | 0.92 ± 0.11   | 0.92 ± 0.11   | 0.92 ± 0.11   | 0.91 ± 0.11  |
>
>   To illustrate the performance drop by pruning from dense transfer, we normalized the average results against dense transfer, as shown in the following table:
>
>   |                          | Dense Transfer | 7:8  | 6:8  | 5:8  | 4:8  | 3:8  | 2:8  | 1:8  |
>   |--------------------------|----------------|------|------|------|------|------|------|------|
>   | Sparse to Dense Transfer | 1.00           | 1.00 | 0.93 | 1.00 | 0.96 | 0.59 | 0.56 | 0.55 |
>   | Sparse Transfer          | 1.00           | 1.00 | 1.00 | 1.00 | 0.93 | 0.95 | 0.88 | 0.68 |
>   | Dense Transfer with Freezing | 1.00      | 0.98 | 1.04 | 1.03 | 1.01 | 1.01 | 1.01 | 1.00 |
>
>   We observed a repeating pattern that is consistent with Figure 6 in our original submission. Specifically, the methods Dense Transfer and Dense Transfer with Freezing significantly outperform the other two methods. Notably, removing small-magnitude weights from pre-trained models leads to a marked decrease in performance.

---

> ### Author Response · Authors · 2023-11-21
> **Response to Reviewer 52Ea （2/2）**
>
> **Questions 3:** *Explain the fluctuations in the figures.*
>
> - This could be attributed to two factors. First, pruning may introduce noise since performance doesn't always show a consistent, monotonic decrease with increased weight pruning, similar results are observed in [1,2].  Secondly, results averaged from limited random seeds might intensify this noise. Interestingly, a similar pattern of fluctuation persists in N:M QNLI sparse transfer even after 20 random seeds are run, specifically sparse transfer 3:8 still outperforms 4:8 in QNLI as shown in the above table.
>
> [1] Jaiswal A, Liu S, Chen T, et al. The Emergence of Essential Sparsity in Large Pre-trained Models: The Weights that Matter. NeurIPS 2023.
>
> [2] Liu S, Chen T, Zhang Z, et al. Sparsity May Cry: Let Us Fail (Current) Sparse Neural Networks Together!. ICLR 2023.

---

### Official Review · Reviewer_ahA2 · 2023-11-01

**Soundness:** 3 good
**Presentation:** 3 good
**Contribution:** 3 good
**Rating:** 6
**Confidence:** 4

**Summary:**

This paper investigates the significance of low-magnitude weights in pre-trained language models and how they affect performance in downstream tasks. The authors suggest a task-centric method to prune pre-trained language models. They illustrate that the small-magnitude weights hold crucial downstream knowledge essential for addressing more difficult tasks, challenging the conventional wisdom regarding the relevance of "Junk DNA" in the human genome and its similarity to the redundancy observed in deep neural networks.

**Strengths:**

This article introduces three novel discoveries that set it apart from prior techniques for pruning Large Language Models (LLMs) such as essential sparsity, WANDA, and SparseGPT:

1. The paper adopts a task-centric viewpoint when considering pre-trained weights, offering a more holistic comprehension of the function of small-magnitude weights in LLMs and their influence on performance, particularly in complex tasks. This viewpoint is innovative and challenges conventional wisdom.

2. The paper mainly employs magnitude-based pruning to identify and measure the subtle importance of low-magnitude weights. While this approach has been used in previous research, the paper introduces a more nuanced and task-specific application of this technique.

3. The paper challenges the established beliefs regarding the role of "Junk DNA" in the human genome and its similarity to the redundancy observed in deep neural networks. By expanding the Junk DNA Hypothesis to encompass other criteria for weight importance, the paper offers a more comprehensive insight into the significance of low-magnitude weights in LLMs and their impact on performance.

**Weaknesses:**

1. This paper does not provide another LLM pruning method. As stated above it is mainly considered as a strength (with its simplicity and great clarity). However, it remains uncertain how the magnitude-based pruning approach would yield practical application value because (1) this vanilla pruning technique leads to a rapid decline in performance, and (2) unstructured sparsity is impractical for GPU implementation.

2. Furthermore, the majority of experiments indicate that pruning performance, even for moderately challenging tasks, begins to drop at medium sparsity (around 30-50%). This raises doubts about the potential for any acceleration in LLM inference speed resulting from such pruning techniques.

**Questions:**

Have the authors examined their study topic for quantization?

---

> ### Author Response · Authors · 2023-11-21
> **Response to Reviewer ahA2**
>
> We thank you for your overall positive comments. We are really grateful that you recognize three novel discoveries of our paper. We have carefully read the review and would like to address them point-by-point as below.
>
>
> **Question 1:** *This paper does not provide another LLM pruning method. As stated above it is mainly considered as a strength (with its simplicity and great clarity). However, it remains uncertain how the magnitude-based pruning approach would yield practical application value because (1) this vanilla pruning technique leads to a rapid decline in performance, and (2) unstructured sparsity is impractical for GPU implementation.*
>
> -  We thank the reviewer for identifying the major strength of our work. As already highlighted by the reviewer, our intent was not to demonstrate a new pruning method, but to present a detailed analysis of the importance of the low magnitude weights from a task-centric angle. For that, we view magnitude pruning as a *quantitative and easily controllable tool* for investigating and understanding the role of small-magnitude weights within pre-trained LLMs. Additionally, we consider magnitude pruning to be a widely used post-training pruning technique since it does not necessarily require more sophisticated proxies such as gradient or Hessian. Also, as the models grow bigger, even fine-tuning for few epochs may not be feasible for a downstream user to evaluate such proxies. We believe magnitude-based proxy has value due to its ease of adaptation without any additional fine-tuning or proxy storage cost.
> -  Please note, we have demonstrated the Junk DNA hypothesis both for unstructured and N:M sparsity. It may often be argued that having inference speed up for unstructured sparsity is difficult without proper hardware or compiler support. However, N:M sparsity can yield inference speed up ([Accelerating Inference with Sparsity Using Ampere and TensorRT](https://developer.nvidia.com/blog/accelerating-inference-with-sparsity-using-ampere-and-tensorrt/)).
>
>
>
> **Question 2:** *Furthermore, the majority of experiments indicate that pruning performance, even for moderately challenging tasks, begins to drop at medium sparsity (around 30-50%). This raises doubts about the potential for any acceleration in LLM inference speed resulting from such pruning techniques.*
>
> - Regarding the reviewer's concern on the potential of pruning for LLMs, we want to highlight that our goal is not to demonstrate a pruning method that can yield speedup. Rather, we take the first step to analyze the impact of pruning from a task-centric point of view. More importantly, we take the first step to answer a rather overlooked question: *whether the low-magnitude weights are truly inconsequential artifacts for large-scale models?*
> - Nevertheless, apart from the detailed analysis on the importance of low-magnitude weights, we believe our discoveries should inspire exploration into potential avenues, such as task-complexity-dependent dynamic inference and network self-slimmable properties. Specifically, we can dynamically assign a different amount of weights according to specific prior knowledge of task difficulty, and then we can implement more precise and hardware-friendly methods to dynamically prune models, thereby facilitating efficient inference.
>
>
> **Question 3:** *Have the authors examined their study topic for quantization?*
>
> - We appreciate the reviewer's suggestion. As you suggested, we explored the Junk DNA Hypothesis in the field of quantization. We use GPTQ [1] to quantize the weights and generate the results with different target quantizations Bit-width. Specifically, we examined with Setting 3: With vs. Without Available External Information. We use Vicuna-7B as the base model and report the results in the following table, the values before the slash ("/") represent Exact Match Ratios, while the values after the slash are Normalized Against Pre-Quantization Performance.
>
>   | Configuration      | Closed Book (hard task)  | Open Book (easy task)  |
>   |--------------------|----------------------------|---------------------------|
>   | Before Quantization| 63.7 / 1.0                 | 76.1 / 1.0                |
>   | 8-bit              | 62.8 / 0.98                | 75.55 / 0.99              |
>   | 4-bit              | 54.35 / 0.85               | 72.2 / 0.95               |
>   | 2-bit              | 13.5 / 0.21                | 18.65 / 0.25              |
>
>
>     Our findings demonstrate a notable trend: as the bit-width decreases, there is a more rapid decline in performance after quantization for hard tasks. This trend also indicates the connection between the Junk DNA hypothesis and quantization.
>
>     [1] Frantar E, Ashkboos S, Hoefler T, et al. Gptq: Accurate post-training quantization for generative pre-trained transformers[J]. arXiv preprint arXiv:2210.17323, 2022.

---

### Meta-Review · Area_Chair_LeJg · 2023-12-10

**Metareview:**

This paper investigates magnitude-based pruning for NLP tasks.

The reviewers all appreciate the paper and all vote for acceptance with 6/6/8 scores. However, this AC has some serious reservations about the paper:

1. The "Junk DNA" framing appearing prominently in the title, abstract and introduction should be removed. The connection to junk DNA is at best contrived and not even a good analogy (DNA: genotype/code versus NN weights: a consequence of model choices and training data).

2. The experiments with freezing low magnitude weights and only updating the large magnitude weights are not very meaningful. Recent results on low rank techniques (LoRA) show that one can change a very small number of parameters and still get good (fine-tuning) performance. So I suspect that one would essentially be able to see the same behaviour with freezing any subset of weights.

3. Magnitude-based pruning is a very simple minded approach. LeCun et al already in 1990 showed that it is possible to get a much more advanced estimate of saliency with little computational overhead.

For these reasons, it is recommended that the paper is reject to give the authors an opportunity to improve their in many other ways excellent contribution.

**Justification For Why Not Higher Score:**

Problems with the overall framing and some of the experiments. If the SAC wants to bump up due to liking the paper and wanting to go with the referees recommendations then I will not protest.

**Justification For Why Not Lower Score:**

None.

---

### Decision · Program_Chairs · 2024-01-16

Reject